# Assessment of Standing Multi-Frequency Bioimpedance Analyzer to Measure Body Composition of the Whole Body and Limbs in Elite Male Wrestlers

**DOI:** 10.3390/ijerph192315807

**Published:** 2022-11-28

**Authors:** Yeong-Kang Lai, Chu-Ying Ho, Chung-Liang Lai, Chih-Yang Taun, Kuen-Chang Hsieh

**Affiliations:** 1College of Electrical Engineering, National Chung Hsing University, Taichung 40227, Taiwan; 2Department of Physical Medicine and Rehabilitation, Puzi Hospital, Ministry of Health and Welfare, Chiayi 61347, Taiwan; 3Department of Occupational Therapy, Asia University, Taichung 41354, Taiwan; 4Department of Exercise Health Science, National Taiwan University of Sport, Taichung 40404, Taiwan; 5Department of Research and Development, Starbia Meditek Co., Ltd., Taichung 40227, Taiwan; 6Big Data Center, National Chung Hsing University, Taichung 40227, Taiwan

**Keywords:** body composition, dual energy X-ray absorptiometry, bioelectrical impedance, athletes

## Abstract

We investigated differences in body composition measurements for the whole body and limb segments in elite male wrestlers between results of multi-frequency bioelectrical impedance analyses (MF_BIA_) and dual energy X-ray absorptiometry (DXA). Sixty-six elite male wrestlers from Taiwan were recruited. Wrestlers’ body fat percentage (PBF_WB_), whole body fat-free mass (FFM_WB_), whole body lean soft tissue mass (LSTM_WB_), and fat-free mass of arms, legs and trunk (FM_Arms_, FFM_Legs_, FFM_Trunk_) were measured by MF_BIA_ and DXA, and analyzed using Pearson correlation coefficient and Bland–Altman plot. Correlations of FFM_WB_, LSTM_WB_, and PBF_WB_ between devices were 0.958, 0.954, and 0.962, respectively. Limits of agreement (LOA) of Bland–Altman plot were −4.523 to 4.683 kg, −4.332 to 4.635 kg and −3.960 to 3.802%, respectively. Correlations of body composition parameters FFM_Arms_, FFM_Legs_ and FFM_Turnk_ between devices in each limb segment were 0.237, 0.809, and 0.929, respectively; LOAs were −2.877 to 2.504 kg, −7.173 to −0.015 kg and −5.710 to 0.777 kg, respectively. Correlation and consistency between the devices are high for FFM, LSTM and PBF but relatively low for limb segment FFM. MF_BIA_ may be an alternative device to DXA for measuring male wrestlers’ total body composition but limb segment results should be used cautiously.

## 1. Introduction

Common wrestling competitions that appear often in Olympic Games and college sports can be divided into freestyle and Greco-Roman style. Wrestling competition requires intense physical activity using the power and strength of the body musculature and the isometric force of various wrestling techniques [1]. The wrestlers’ strength, speed, and dynamics all affect the success rate of wrestling. Due to the need for short-term and high-intensity performance, anaerobic strength and ability are crucial in wrestling competition [2].

Varddar et al. [3] pointed out that changes in the maximum power of excellent wrestlers is related to the wrestlers’ fat-free mass (FFM). That is, the average power and minimum power correlate significantly with the wrestlers’ FFM. Body fat comes from adipose cells and lipids in body tissues. Most wrestlers try to lose weight, especially percent body fat (PBF), because their actual weight must match their weight class before competition. Therefore, optimal body composition is one of the key points wrestlers must focus on. Wrestlers and coaches believe that PBF is a body composition factor that must be controlled, and that a lower PBF is beneficial to the athlete’s competitive performance. Numerous studies have shown that FFM is a more reliable predictor of wrestlers’ anaerobic performance, rather than PBF [4,5]. Improper weight control or diet will reduce protein nutrition, thereby impairing muscle function and reducing FFM [6].

The assessment of FFM or lean soft tissue mass (LSTM) measurements of the whole body and each limb segment has wide availability and utility in the sports performance industry [7]. In particular, the LSTM of each limb segment is important to the athlete’s performance in different sports, as well as the assessment of injury risk. Numerous studies have described the specific needs and relationships of professional and collegiate male and female athletes in different sports, positions, and total and regional body composition [8,9,10,11]. Further research on this potential relationship is warranted. In addition, studies have also found that, from pre-season to mid-season, the LSTM of the legs of elite hockey players increased, while the PBF of the players decreased [12]. Additionally, about 60% of the weight gain of elite football players during the season comes from an increase in trunk and leg LSTM results [13]. These studies point to the importance of monitoring changes in whole body composition over time or between seasons, but measurements of segmental or local body composition changes may be more informative, especially for different sports, task positions or competitive sports of different classes. Knowing the FFM or LSTM of a player’s body composition in the whole body and segmentally is particularly meaningful in designing the athlete’s training plans [14].

Several ways are available to indirectly assess body composition throughout the body, including body mass index (BMI), waist-to-hip ratio, and skin-fold measurements in anthropometrics [15,16,17]. Computed tomography (CT) and magnetic resonance imaging (MRI) are similar in measuring human body composition. Both methods use cross-sectional tomography to measure the area of muscle tissue, fat tissue, and organs in a single image, and use multi-slice image scanning to measure the volume of different tissues or organs through three-dimensional reconstruction. DXA regards the human body as a three-components model, including fat, lean tissue and bone mineral. These three components are sensitive to the different degree of photon absorption. From the attenuation of the photon beam irradiated in the human body, the tissue composition in the body can be judged. With its excellent accuracy, applicability and relatively low radioactivity, DXA has become the main method for the diagnosis of osteoporosis and the measurement of body composition. It plays an important role in clinical and epidemiological investigations and research [18]. If more accurate measurements of body composition are required, non-invasive methods such as computed tomography (CT) or tomographic techniques such as MRI [19,20], or dual-energy X-ray absorptiometry (DXA) [21] are available options. The high cost and time-consuming nature of CT and MRI are the main limitations of their wider application. Although CT involves the use of radiation, these reference methods provide accurate measurements when performing precise pathological examinations or research. Compared with MRI or CT, the application cost of DXA for whole body measurement and local body composition measurement is relatively low. However, because DXA requires the limitation of a specific field and a specific operator, it cannot actually be used for a large number of cases or long-term body composition measurements in athletes.

Bioelectrical impedance analysis (BIA) is a simple, safe, rapid, and non-invasive method for assessing body composition. Many studies have compared the measurement results of the reference method with BIA, which has been widely used in clinical and epidemiological studies [22]. BIA can also be used to measure athletes’ body composition, primarily using multifrequency and multi-limb segment measurement modes [23]. InBody770 from South Korea has measurement frequencies of 1, 5, 50, 250, 500, 1000 KHz. The body composition of the subject is estimated by the resistance and reactance measured by the weak current of the whole body and each limb segment, in conjunction with the body measurement parameters. This device has been widely used in the measurement of body composition of children, older adults, and the general population and has been validated in several application and validation studies [24,25,26,27,28] and an ambulatory population [29], including athletic young adults [30], football players [31], college athletes [32], soccer players [33] and other groups. The related research on the accuracy of the measurement results was limited. Although there were also professional, young athletes in multi-frequency bioelectrical impedance analyses (MF_BIA_)-related verification studies have been proposed, but most of them were American football, basketball, and other ball games. In wrestling, boxing and other martial arts sports, the verification of research on the percentage of body fat or fat-free mass has been published [34]. However, the six-frequency MF_BIA_ verification research on the body composition or lean mass of each limb was very limited. Therefore, we assume that InBody770 is used as the DXA device for measuring and calibrating the limb lean mass of excellent wrestlers in Taiwan, and there is interchangeability between the two measurement results. However, verification of body composition measurement results (whole body and segmental) of wrestlers using multi-frequency and multi-limb segment bioimpedance analysis is lacking. Therefore, this study aims to investigate differences in body composition measurements for the whole body and limb segments in elite male wrestlers of different weight classes between results of MF_BIA_ and DXA.

## 2. Materials and Methods

### 2.1. Participants

This prospective cross-sectional verification study recruited 66 active-duty male wrestlers at the highest level (elite status) in Taiwan. This study was conducted during off-season. The subjects had received at least eight years of professional training in wrestling sports. For subjects with kidney disease or metabolic syndrome and subjects whose body weight changed by more than 6 kg within three months, have been excluded from this study. In addition to prohibiting strenuous exercise 24 h before the test, subjects did not drink alcoholic beverages or take diuretics and other drugs within one week before the test to control the hydration status of the subjects. Thirty minutes before the BIA measurement, the subjects emptied their bladders with urine. Among them, 23 players had participated in the Asian Games or wrestling competitions at the same level within two years before this experiment. Data were collected during non-competition or seasonal periods from May to August each year (2017 to 2021). All players performed routine training 4–5 times per week during the study period. All players were aged 18 years or older.

### 2.2. Ethical Considerations

The Human Experimentation Committee of the Caotun Psychiatric Center of the Ministry of Health and Welfare reviewed and approved the study protocol (IRB-104051, IRB-108032). After receiving an explanation and fully understanding the experimental purpose of the study and its procedures and potential risks, all subjects provided signed informed consent prior to participating in the study. The coefficient of variation of impedance measurements of the right hand-to-foot current flow path was assessed using within-day and between days measurements. Five male and five female participants repeated the impedance measurement 10 times within one hour of the day. Between days impedance measurements were performed within the same time period over five days.

### 2.3. Procedure

The subjects first measured their body parameters such as height, weight and filled out training and health questionnaires. Then, the subjects used a bioimpedance analyzer and DXA to measure body composition of the whole body and each limb segment. Subjects were measured after adequate rest, four hours of fasting, and adequate hydration prior to the experiment. They were instructed to follow a standard dietary regimen, avoid strenuous exercise, and to avoid stimulants and depressants for 24 h prior to the test. They were advised to remove all metal objects and trims and wear light clothing during measurement. The procedure was completed within two hours. The testing environment included a temperature- and humidity-controlled room. The subject’s weight was measured with MF_BIA_-InBody770, and the weight measurement was accurate to 0.1 kg. The subject was measured with a height ruler (Holtain, Cosswell, Wales, UK) without shoes. Height measurement was accurate to 0.5 cm. Height and weight measurements were performed twice with the average value taken.

### 2.4. Bioimpedance Analysis

In this study, eight-pole bioimpedance body composition analyzer InBody770 (Biospace Co., Ltd., Seoul, Republic of Korea, MF_BIA_) was used. Ten minutes before the MF_BIA_ measurement, subjects assumed a standing position for 10 min to stabilize body fluids. Arms were kept straightened during the test to prevent them from touching the inner thigh. All four fingers of the subject’s left and right hands were in contact with the grip electrode surface, the thumb was in contact with the oval shock surface, and the heels were aligned with the edge of the foot shock. The subject stood barefoot on the shock. The contact method between the testing posture and the electrode plate followed the instruction in the operation manual, and the hands and feet were moistened with an electrolyte-saturated paper towel before the test. Each subject was measured twice, and results were averaged, with an interval of no more than three minutes.

### 2.5. Dual-Energy X-ray Absorptiometry

DXA (Lunar prodigy; GE medical System, Madison, WI, USA) was performed after MF_BIA_ measurements were recorded. The measurement time did not exceed 15 min. Subjects lay supine on the test table, and the scan was performed from the head to the feet to measure lean mass, fat mass, and bone mineral content. Encore 2003 Version 7.0 analytical software (Encore Analytics, Tacoma, WA, USA) was used to analyze whole body lean mass (Lean_WB_), fat mass (FM_WB_), bone mineral mass (BMC_WB_), body fat percentage (PBF_WB_), right upper extremity fat-free mass (FFM_RA_), left upper extremity fat mass (FFM_LA_), body fat-free mass (FFM_TK_), right lower limb fat-free mass (FFM_RL_), left lower limb fat-free mass (FFM_LL_), upper limb fat-free mass (FFM_Arms_), lower limb fat-free mass (FFM_Legs_).

### 2.6. Statistical Analysis

Data are presented as mean ± SD. The minimum and maximum values are shown in parentheses. Continuous variables in this study include age, weight, height, and BMI and all of them were normally distributed according to the Shapiro–Wilk tests. Correlation analysis and paired-*t* test were performed on the whole body FFM, LSTM, BFM, PBF, and the FFM of each limb segment by BIA and DXA. Lin’s CCC (concordance correlation coefficient) was applied to evaluate the consistency of the two methods. When CCC < 0.90, the agreement was interpreted as poor, 0.90–0.95 was moderate, 0.95–0.99 was substantial, and above 0.99 was perfect. Accuracy of the measurement was expressed as a bias correction factor (C_b_). Results measured by DXA were used as the standard. A Bland–Altman Plot was used to display the mean difference (bias) between the two methods, the upper and lower bounds of agreement, and the regression line equation for the corresponding trend. For FFM measurement results of the two devices on the upper and lower limbs, the y-axis of the Bland–Altman Plot was calculated as the percentage of the difference between the two methods. All statistical analyses were performed using the software SPSS Version 20 (IBM SPSS, Armonk, NY, USA) and MedCalc Version 16.0 (MadCalc Software, Mariakerke, Belgium). A value of *p* < 0.05 was established as statistical significance. A Power analysis was conducted for detecting difference in lean soft tissue mass (LSTM) which revealed a sample size requirement of 55.

## 3. Results

A total of 66 elite male wrestlers were recruited for this experiment, including 32 Freestyle and 34 Greco-Roman. Based on wrestlers’ weights, they were divided into three weight classes: lightweight, *n* = 12; middleweight, *n* = 40; and heavyweight, *n* = 14. Mean age was 20.6 ± 1.1 years, mean weight was 76.4 ± 16.3 kg, mean height was 170.6 ± 6.3 cm, mean body mass index was 26.1 ± 4.3 kg/m^2^, mean PBF was 17.9 ± 9.2%, and mean professional training time was 8.5 ± 2.1 years. PBF and BMI increased with the increase in magnitude. Subjects’ baseline characteristics are shown in Table 1.

In Table 2, results for FFM, LSTM, BFM and PBF of the DXA and MF_BIA_ devices are shown for the whole body, right arm (RA), left arm (LA), trunk (TK), right leg (RL), left leg (LL), upper limb (Arms), and lower limb (Legs). Results of Pearson correction coefficient, bias correction factor, concordance correlation coefficient (CCC), and paired-*t* test are also presented. Results of the two devices were calculated using the mean difference (bias) of the Bland–Altman Plot, limits of agreement (LOA), lower limits (−1.96 SD), upper limits (upper limits, +1.96 SD), and the regression line in the Bland–Altman plot showing agreement and difference. The CCC values of FFM_WB_, LSTM_WB_, BFM_WB_, and PFB_WB_ for the whole body ranged from 0.938 to 0.979, showing excellent consistency, reliability of repeatability, and the measurement results of the whole body composition between the two devices. However, the value of CCC of each limb segment was between 0.233 and 0.440, except that the FFM_TK_ was slightly higher at 0.755. Consistency of the FFM measurements representing the two devices at each limb segment was questionable. The mean differences of FFM_WB_, LSTM_WB_, and BFM_WB_ in the Bland–Altman plot of the whole-body measurement results of the two devices were 0.077, 0.151, and −0.07 kg, respectively, indicating excellent consistency.

Figure 1a is the distribution diagram of the two devices in FFM_Arms_. The red solid line is the equivalent line, and the blue solid line is the regression. The correlation between the measurements of the FFM_Arms_ of the two devices was r = 0.237. Figure 1b was the Bland–Altman plot of the two devices at FFM_Arms_. The x-axis is the average of the two devices, and the y-axis is the difference between the two measurements divided by the DXA observations multiplied by 100%. In the figure, the LOA of MF_BIA_ at FFM_Arms_ ranged from −38.6 to 33.1%.

Figure 2a is the distribution diagram of the two devices in FFM_Legs_. The correlation between the measurements of the two measuring devices was r = 0.655. Figure 2b was a Bland–Altman plot of the two devices, which showed that the LOA of MF_BIA_ in FFM_Legs_ ranged from −33.0 to −1.9%.

## 4. Discussion

This study is the first to investigate the body composition measurement results of the whole body and limbs of male wrestlers using MF_BIA_ compared with results of DXA. Results showed that MF_BIA_ applied to body composition measurement results of male wrestlers had good interchangeability compared with the measurement results of the DXA reference. However, a certain deviation was noted in the measurement results of FFM of each limb segment, and correlations were not as good as those for the whole body.

DXA belongs to low radiation dose. The effective dose of adult whole body examination is 0.1 to 75 μSv, which is equivalent to the equivalent dose of the subject’s natural background radiation for one day or less [35].

In previous evaluations of the InBody770, Brewer et al. [32] used DXA to compare the FFM measurement results of InBody770 in legs and arms among 43 American Division I college male athletes, and their biases were −6.57 and −1.30 kg, respectively. Those authors, therefore, did not recommend the use of InBody770 for body composition measurement in athletes. Esco et al. [36] used DXA to compare InBody720 body composition measurements of the whole body and each limb for 45 female college athletes. For whole PBF, FFM, and lean mass, the correlation coefficients of the two devices were r = 0.94, 0.95, 0.92, respectively. For the FFM of the upper and lower limbs, the correlation coefficients of the two devices were r = 0.89, 0.83, respectively. Those authors concluded that the InBody720 was a good alternative to DXA for measuring body composition of the whole body and limbs of female athletes. Additionally, Brewer et al. [24] applied the four-compartment model criterion to compare InBody770 with whole PBF in 82 healthy young adults, and the biases of male and female subjects were overestimated by 2.2% and 4.0%, respectively. Although the error of female subjects was larger, the authors concluded that InBody770 had good accuracy in determining the PBF of healthy young people.

In the present study, the measurement results for the whole body in terms of correlations and consistency were better than the measurement results for each limb segment. To further explore the reasons for this discrepancy, the subjects from this study were limited to males, aged between 19 and 23 years, with a BMI of 21 to 38 kg/m^2^ and a mean of 26.1 ± 4.3 kg/m^2^. PBF ranged from 5% to 37%, with a mean of 17.9 ± 9.2%. In terms of age, the subjects of this study were young and normal distribution. It has been pointed out in many studies that many BIA FFM estimation equations could be omitted for estimating the age variable, especially for athletes [37]. An outcome factor affecting the MF_BIA_ whole body measurement is whether the built-in equations can predict the subject’s body composition through the estimated variables. For example, in another study, the mean BMI and PBF of 20-year-old male Asians were 23.0 ± 3.5 kg/m^2^ and 21.2 ± 6.9%, respectively [38].

We reasoned that the subjects of the present study were healthy and strong wrestlers. Therefore, the body composition measurement results in the whole body should be more in line with sampling objects of the estimation model established by MF_BIA_. In general, manufacturers of body composition measuring devices treat the built-in body composition equation as a trade secret. The sampling objects required to establish the equation must take the diversity of future users into account. Additionally, when recruiting subjects, it may be necessary for them to be relatively healthy and biased towards normal distribution. Therefore, in the evaluation and validation of many MF_BIA_ or other BIA devices, the object of discussion often determines the validation results. Generally speaking, unhealthy subjects or subjects at both ends of the normal distribution curve of the population, MF_BIA_ or BIA devices often have poor verification results. This is the basis for establishing the BIA body composition equation, and also explains the large deviations among measurement errors.

Although the MF_BIA_ discussed in the present study has sufficient reference value in the measurement of body composition of male wrestlers in the whole body, the reference value of the FFM in each segment, especially the limbs, was limited in the measurement results of MF_BIA_. To explain this limitation, we deduced that wrestler’ bodies, especially the proportion of lean body mass in the limbs, is different from that of healthy men of the same age. The appendicular lean mass index (ALMI) of the subjects in the present study was 9.62 ± 1.32 kg/m^2^. White and Mexican American males in the same age group had mean ALMI results of 8.87 ± 1.34 and 8.56 ± 1.05 kg/m^2^, respectively [39]. When applying MF_BIA_ to measure the body composition of each limb segment of elite wrestlers, the reference value may be limited.

Whether MF_BIA_, BIA commercial devices or BIA body composition estimation equations are published in academic journals, they are all estimation models established by using experiments to collect estimated variables and response variables. The body composition estimation equation applied to the subject usually has a certain equation establishment procedure. The sampling source and distribution of the subjects are often limited by funding. If the sampling object is limited and lacks repeated verification, its application value will be limited. For commercial devices, the information related to the built-in equations is often a trade secret or a black box. Under the conditions of establishing the above equations, users with special physical or physiological conditions cannot judge the reference value of the measurement results when using commercial BIA to measure body composition. Therefore, in line with the purpose of the present study, extensive validation studies are required for commercial BIA devices to increase their practical value.

Whether multi-frequency and multi-limb segments are superior to traditional single-frequency, whole-body bioimpedance analysis in body composition measurement still needs further and a large number of verification studies to clarify. MF_BIA_ is used in body composition measurement because of its measurement variables. The body composition items that can be measured by the MF_BIA_ device also increase a lot. The accuracy of the body composition measurement results often varies with the device manufacturer, the nutritional status of the testee, age, phenotype, and competition intensity. Using MF_BIA_ can easily obtain these measurement results, but it needs to be cautious when it is actually used in clinical practice or training. Furthermore, it is necessary to abide by its measurement specifications and restrictions in order to obtain more valuable estimation results.

In many sports, especially weight-based competitions, participants often suffer from acute or chronic dehydration and energy intake restriction. Weight control in abnormal ways can affect the health of athletes, especially young athletes Several studies have shown that rapid weight control is common in high school or international wrestling competitions [40,41]. This is a very important issue for body composition measurement. In this study, because the test period was off-season, the subjects were not undergoing acute or chronic dehydration or weight control at the time of the test. Compared with body composition measurement methods of image scanning, such as DXA or CT, bioimpedance analysis has more assumptions and requirements for body composition measurement. When bioimpedance analysis is used for single-frequency or whole-body body composition measurement, The weight of the subject needs to be input or measured. However, in the BIA measurement of multiple segments, it is difficult to confirm or measure the weight of the upper or lower limbs. The current situation can only be estimated. This affects the upper or lower limbs LMST estimate is accurate

This study attempted to investigate the consistency of DXA and MF_BIA_ in measuring the parameters of total body composition and each limb segment, using excellent young wrestlers from Taiwan as the subjects. The findings do not apply to female wrestlers, nor do they apply to other races, such as white or black wrestlers. As MF_BIA_ becomes more widely used for body composition measurement in athletes, the validity of MF_BIA_ device measurements must be determined. The results of this study showed that for young male wrestlers, it is necessary to establish the corresponding fat-free mass or LSTM estimation equations for each limb segment, and to verify the validity and measurement accuracy of the device used. According to research results, InBody770 was used for the lean mass of men’s upper and lower limbs, and the measurement results of DXA were not interchangeable.

## 5. Conclusions

MF_BIA_ body composition measurements for the whole body are consistent in male wrestlers, but significant differences are found in FFM measurements of limb segments. MF_BIA_ may be an alternative device to DXA for measuring male wrestlers’ total body composition, but limb segment results should be used cautiously.

## Figures and Tables

**Figure 1 ijerph-19-15807-f001:**
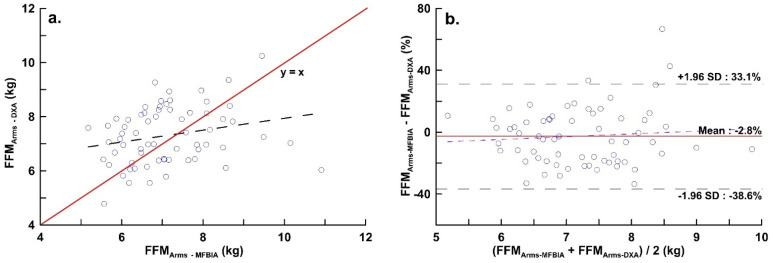
(**a**) Distribution of upper limbs fat-free mass measured by DXA and MF_BIA_ (y = 5.749 + 0.219x, r = 0.237, Residual Stand Deviation = 1.043, *p* = 0.055); (**b**) Bland–Altman Plots of upper limbs fat-free mass measured by DXA and MF_BIA_ (bias: −2.755%, LOA: −38.64% to 33.12%, y = −14.637 + 1.644x, *p* = 0.530). The red line and the dotted line represent the bias and LOA respectively.

**Figure 2 ijerph-19-15807-f002:**
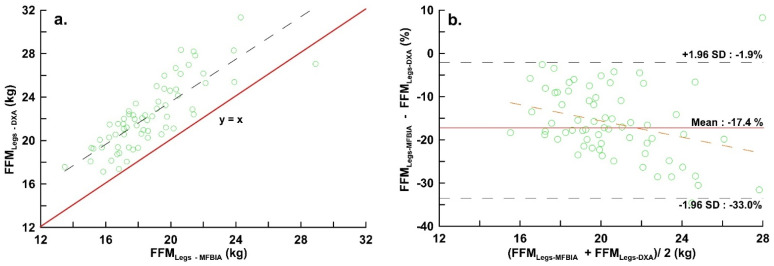
(**a**) Distribution of lower limbs fat-free mass measured by DXA and MF_BIA_ (y = 3.984 + 0.979x, r = 0.655, Residual standard deviation = 1.839, *p* < 0.001; (**b**) Bland–Altman Plots of lower limbs fat free mass measured by DXA and MF_BIA_ (bias: −17.44%, LOA: −33.02% to −1.87%, y = −12.843 − 0.224x, *p* = 0.366). The red line and the dotted line represent the bias and LOA respectively.

**Table 1 ijerph-19-15807-t001:** Subjects’ characteristics of elite wrestlers in the three weight classes.

	Light Weight (*n* = 12)	Middle Weight (*n* = 40)	Heavy Weight (*n* = 14)	All (*n* = 66)
Age (year)	20.1 ± 1.1 (18.7, 21.1)	20.6 ± 1.1 (18.7, 22.3)	21.1 ± 1.0 (19.4, 22.3)	20.6 ± 1.1 (18.7, 22.3)
Weight (kg)	60.6 ± 2.7 (56.4, 63.9)	71.8 ± 5.6 (64.1, 83.8)	103.1 ± 13.0 (84.0, 127.9)	76.4 ± 16.3 (56.4, 127.9)
Height (cm)	164.2 ± 4.5 (156.5, 171.0)	170.4 ± 4.3 (162.0, 181.0)	176.7 ± 7.0 (170.0, 196.0)	170.6 ± 6.3 (156.5, 196.0)
BMI (kg/m^2^)	22.5 ± 1.2 (20.7, 24.5)	24.8 ± 2.1 (20.9, 29.1)	33.0 ± 3.2 (27.5, 37.5)	26.1 ± 4.3 (20.7,37.5)
Percent Body Fat (%)	10.7 ± 3.7 (5.1, 17.7)	15.7 ± 7.2 (8.1, 36.2)	30.3 ± 5.0 (21.8, 37.1)	17.9 ± 9.2 (5.1, 37.1)
FFM (kg)	54.5 ± 2.9 (50.2, 60.8)	60.8 ± 4.8 (49.4, 74.3)	71.4 ± 6.1 (65.0, 86.1)	61.9 ± 7.3 (49.4, 86.1)
LSTM (kg)	51.4 ± 2.7 (47.6, 57.3)	57.3 ± 4.9 (46.1, 70.4)	67.5 ± 6.1 (60.8, 82.2)	58.4 ± 7.0 (46.1, 82.2)
TBW (kg)	40.2 ± 2.1 (36.9, 44.8)	44.8 ± 3.5 (36.4, 54.8)	52.6 ± 4.5 (47.9, 63.4)	43.0 ± 5.4 (36.4, 63.4)
Training experience (year)	7.9 ± 1.9 (6.7, 9.3)	8.2 ± 1.6 (6.9, 8.9)	8.8 ± 1.3 (5.9, 9.5)	8.5 ± 2.1 (6.7, 9.5)

BMI: Body Mass Index; FFM: Fat-free mass; LSTM: Lean soft tissue mass; TBW: Total body water.

**Table 2 ijerph-19-15807-t002:** Correlation of body composition estimates using Pearson product moment correlation and Bland–Altman plot.

							Limit of Agreement				
	MF_BIA_	DXA	*ρ*	CCC	*p* ^1^	C_b_	Bias	Lower	Upper	Regression Line	Trend
FFM_WB_ (kg)	61.97 ± 8.10	61.89 ± 7.27	0.958	0.953	0.790	0.994	0.077	−4.523	4.683	Y = −6.956 + 0.109x	**
LSTM_WB_ (kg)	58.55 ± 7.59	58.40 ± 7.02	0.954	0.951	0.519	0.997	0.151	−4.332	4.635	Y = −4.475 + 0.079x	*
BFM_WB_ (kg)	14.02 ± 9.47	14.09 ± 10.16	0.982	0.979	0.772	0.997	−0.07	−3.963	3.802	Y = 0.930 − 0.710x	**
PBF_WB_ (%)	17.11 ± 7.51	17.91 ± 9.20	0.962	0.938	*	0.975	−0.795	−6.362	4.771	Y = 2.822 − 0.206x	***
FFM_RA_ (kg)	3.58 ± 0.58	3.69 ± 0.57	0.238	0.233	0.186	0.979	−0.117	−1.526	1.285	Y = −0.268 + 0.041x	0.834
FFM_LA_ (kg)	3.54 ± 0.57	3.61 ± 0.53	0.224	0.221	0.420	0.990	−0.069	−1.424	1.283	Y = −0.439 + 0.103x	0.604
FFM_TK_ (kg)	27.36 ± 3.47	29.83 ± 4.27	0.929	0.755	***	0.813	−2.467	−5.714	0.777	Y = 3.689 − 0.215x	***
FFM_RL_ (kg)	9.39 ± 1.30	11.19 ± 1.61	0.795	0.440	***	0.554	−1.796	−3.715	0.121	Y = 0.645 − 2.372x	***
FFM_LL_ (kg)	9.32 ± 1.27	11.12 ± 2.32	0.811	0.434	***	0.536	−1.798	−3.546	−0.054	Y = 0.247 − 0.200x	*
FFM_Arms_ (kg)	7.12 ± 1.15	7.31 ± 1.06	0.237	0.233	0.273	0.983	−0.186	−2.877	2.504	Y = −1.129 + 0.130x	0.506
FFM_Legs_ (kg)	18.71 ± 2.56	22.30 ± 3.10	0.809	0.440	***	0.544	−3.594	−7.173	−0.015	Y = 0.705 − 0.209x	*

*ρ*: Pearson correction coefficient; C_b_: Bias correction factor; CCC: Concordance correlation coefficient; *: *p* < 0.05, **: *p* < 0.01, ***: *p* < 0.001. FFM: fat-free mass; LSTM: Lean soft tissue mass; BFM: body fat mass; *p*
^1^: Paired samples *t*-test of two-tailed probability. subscript: WB, RA, LA, TK, RL, LL, arms, legs: whole body, right arm, left arm, trunk, right leg, left leg, upper limbs, lower limbs.

## Data Availability

The datasets generated and/or analyzed during the current study are available from the corresponding author on reasonable request.

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
