# Peer review of "Assessment of Standing Multi-Frequency Bioimpedance Analyzer to Measure Body Composition of the Whole Body and Limbs in Elite Male Wrestlers"

_ijerph, 2022, doi:10.3390/ijerph192315807_

Round 1

Reviewer 1 Report

The aim of this study was to investigate differences in body composition measurements for the whole body and limb segments in elite male wrestlers of different weight classes between results of BIA and DXA.

I have some comment to:

- the Intoduction: description for DXA is missing. I am not sure for what purpose CT and MRI are mentioned as they are not used in the study. I propose to add the principle of operation of the devices used and the differences between them.

- Materials and Methods must be improved: what were the inclusion / exclusion criteria from the study? How were height and weight measured? How many times have the measurements been repeated? In the BIA test, the adequate hydration before the measurement is very important. How was it controlled? What were the TBW results?

Statistical tests require examination of the normality of data distribution and appropriate selection of methods. A request to the authors for a supplement. 

Author Response

Reviewer 1.

The aim of this study was to investigate differences in body composition measurements for the whole body and limb segments in elite male wrestlers of different weight classes between results of BIA and DXA.

I have some comment to:

- the Introduction: description for DXA is missing. I am not sure for what purpose CT and MRI are mentioned as they are not used in the study. I propose to add the principle of operation of the devices used and the differences between them.

Response:

Thanks for your suggestion. "Computed tomography (CT) and magnetic resonance imaging (MRI) are similar in measuring human body composition. Both methods use cross-sectional tomography to measure the area of muscle tissue, fat tissue, and organs in a single image, and use multi-slice image scanning to measure the volume of different tissues or organs through three-dimensional reconstruction. DXA regards the human body as a three-components model, including fat, lean tissue and bone mineral. These three components are sensitive to the different degree of photon absorption. From the attenuation of the photon beam irradiated in the human body, the tissue composition in the body can be judged. With its excellent accuracy, applicability and relatively low radioactivity, DXA has become the main method for the diagnosis of osteoporosis and the measurement of body composition. It plays an important role in clinical and epidemiological investigations and research [18]". This narrative has been added to the Introduction Section, Line 72-83.

  1. Ellis KJ. Human body composition: in vivo methods. Physiol. Rev. 2000, 80, 649-680.

- Materials and Methods must be improved: what were the inclusion / exclusion criteria from the study? How were height and weight measured? How many times have the measurements been repeated? In the BIA test, the adequate hydration before the measurement is very important. How was it controlled? What were the TBW results?

Response:

 Thank you for your suggestion. "The subject's weight was measured with MFBIA-InBody770, and the weight measurement was accurate to 0.1kg. The subject was measured with a height ruler (Holtain, Cosswell, Wales, UK) without shoes.. Height measurement was accurate to 0.5 cm. Height and weight measurements were performed twice with the average value taken.”

"This study was conducted during off-season. The subjects had received at least eight years of professional training in wrestling sports. For subjects with kidney disease or metabolic syndrome, and subjects whose body weight changed by more than 6 kg within three months, have been excluded from this study. In addition to prohibiting strenuous exercise 24 hours before the test, subjects did not drink alcoholic beverages or take diuretics and other drugs within one week before the test to control the hydration status of the subjects. Thirty minutes before the BIA measurement, the subjects emptied their bladders with urine. “

“In this study, MFBIA was also used to measure the total water content (TBW) of the subjects, the results were shown in Table 1". These two paragraphs have been respectively supplemented in the Methods section and Results section.

Statistical tests require examination of the normality of data distribution and appropriate selection of methods. A request to the authors for a supplement. 

Response:

Thank you for your suggestion. "Continuous variables in this study include age, weight, height, and BMI and all of them were normally distributed according to the Shapiro-Wilk tests." This description has been added to the statistics section, line 191-192.

Reviewer 2 Report

An original article was made titled:  Standing Multi-Frequency Bioimpedance Analyzer to Measure 2 Body Composition of the Whole Body and Limbs in Elite Male 3 Wrestlers. The study aims to investigate differences in body composition measurements for the whole body and limb segments in elite male wrestlers of different weight classes between results of MFBIA and DXA. Comments and suggestions to strengthen the manuscript are presented below.

1.  Title: The title of your manuscript should identify type of article

2. Abstract: the abstract is ok.

3.  Keywords: I suggest that the keywords be searched in the Mesh database (https://www.ncbi.nlm.nih.gov/mesh/)

4. The introduction was clear. The current state of the research field was reviewed, and key publications cited, and the main aim of the work was mentioned. However, the research question or research hypothesis is not explicitly presented. I suggest adding at least one of them. On the other hand, it is not clear why they want to validate the use of BIA (In-Body 770) in wrestlers if it has already been validated in different sports and athletes.

5.  Methods: They was described with enough detail to allow others to replicate and build on published results. However, some points are not clear: What were the inclusion and exclusion criteria? Was there any fighter excluded from the study?

     The type of sample and sampling used is not indicated. In addition, it does not indicate how the sample size was calculated.

   In addition, it is suggested to follow the STROBE guidelines for cross-sectional studies ((https://www.equator-network.org/reporting-guidelines/strobe/ )

6. Results: Provide a detailed and precise description of the results of study. In addition, it is organized according to the variables analyzed. it is very clear. 

7. Discussion: Authors discussed the results and how they can be interpreted in perspective of previous studies. Although, little is said about their implications or limitations. I suggest adding a section that explicitly accounts for possible clinical implications and practical contributions of yours results. In addition, the limitations section should be more detailed. for example, the amount of the sample, or because the results were not adjusted for co-variables such as nutritional status, phenotype, age, level.

Author Response

Reviewer 2

An original article was made titled:  Standing Multi-Frequency Bioimpedance Analyzer to Measure 2 Body Composition of the Whole Body and Limbs in Elite Male 3 Wrestlers. The study aims to investigate differences in body composition measurements for the whole body and limb segments in elite male wrestlers of different weight classes between results of MFBIA and DXA. Comments and suggestions to strengthen the manuscript are presented below.

Response:

Thank you for your review and suggestions. We will try our best to revise the manuscript according to your suggestions to make it perfect.

  1. Title: The title of your manuscript should identify type of article

Response:

Thank you for your suggestion, we have changed the title of the article to: Assessment of standing multi-frequency bioimpedance analyzer to measure body composition of the whole body and limbs in elite male wrestlers

  1. Abstract: the abstract is ok.
  2. Keywords: I suggest that the keywords be searched in the Mesh database (https://www.ncbi.nlm.nih.gov/mesh/)

Response:

Thank you for your suggestions. According to the content of the Mesh database, we revised the keywords of this article to "Body composition; dual energy X-ray absorption; bioelectrical impedance; athletes".

  1. The introduction was clear. The current state of the research field was reviewed, and key publications cited, and the main aim of the work was mentioned. However, the research question or research hypothesis is not explicitly presented. I suggest adding at least one of them. On the other hand, it is not clear why they want to validate the use of BIA (In-Body 770) in wrestlers if it has already been validated in different sports and athletes.

Response:

Thank you for your suggestion. " The related research on the accuracy of the measurement results was limited. Although there were also professional, young athletes in MFBIA-related verification studies have been proposed, but most of them were American football, basketball and other ball games. In wrestling, boxing and other martial arts sports, the verification of research on the percentage of body fat or fat-free mass has been published [34]. However, the six-frequency MFBIA verification research on the body composition or lean mass of each limb was very limited. Therefore, we assume that InBody770 is used as the DXA device for measuring and calibrating the limb lean mass of excellent wrestlers in Taiwan, and there is interchangeability between the two measurement results”. This narrative has been added to the Introduction section, line 107-11.

"According to research results, InBody770 was used for the lean mass of men's upper and lower limbs, and the measurement results of DXA were not interchangeable." This description will be added to the Discussion section, line 371-373.

  1. Utter A, Lambeth PG. Evaluation of multifrequency bioelectrical impedance analysis in assessing body composition of wrestlers. Medicine &Science in Sports & Exercise. 010 Vol.42 No.2 pp.361-367

  1. Methods: They was described with enough detail to allow others to replicate and build on published results. However, some points are not clear: What were the inclusion and exclusion criteria? Was there any fighter excluded from the study?

Response:

Thank you for your suggestions. " This study was conducted during off-season. The subjects had at least eight years of professional training in wrestling sports. Those with kidney disease or metabolic syndrome and whose body weight changed more than six kilograms within three months were also excluded from the this research". This narrative has been added to the method section, line 124-128.

     The type of sample and sampling used is not indicated. In addition, it does not indicate how the sample size was calculated.

Response:

Thank you for your suggestions.  "A Power analysis was conducted for detecting difference in lean soft tissue mass (LSTM) which revealed a sample size requirement of 55.". This paragraph has been added to the statistical analysis.

  In addition, it is suggested to follow the STROBE guidelines for cross-sectional studies ((https://www.equator-network.org/reporting-guidelines/strobe/ )

Response:

Thank you for your suggestions.  We have checked the 22 suggestions one by one according to the guidelines provided on the above website, and all of them meet the requirements of STROBE's suggestion.

  1. Results: Provide a detailed and precise description of the results of study. In addition, it is organized according to the variables analyzed. it is very clear. 

Response:

Thank you for your suggestions.

  1. Discussion: Authors discussed the results and how they can be interpreted in perspective of previous studies. Although, little is said about their implications or limitations. I suggest adding a section that explicitly accounts for possible clinical implications and practical contributions of yours results. In addition, the limitations section should be more detailed. for example, the amount of the sample, or because the results were not adjusted for co-variables such as nutritional status, phenotype, age, level.

Response:

Thank you for your suggestions. "Whether multi-frequency and multi-limb segments are superior to traditional single-frequency, whole-body bioimpedance analysis in body composition measurement still needs further and a large number of verification studies to clarify. MFBIA is used in body composition measurement because of its measurement variables. The body composition items that can be measured by the MFBIA device also increase a lot. The accuracy of the body composition measurement results often varies with the device manufacturer, the nutritional status of the testee, age, phenotype, and competition intensity. Using MFBIA can easily obtain these measurement results, but it needs to be cautious when it is actually used in clinical practice or training. Furthermore, it is necessary to abide by its measurement specifications and restrictions in order to obtain more valuable estimation results.” . This account has been added to the discussion section, , LINE 339-348.

Reviewer 3 Report

General comments

This manuscript aims at investigating differences in body composition measurements for the whole body and limb segments in elite male wrestlers of different weight classes between results of multi-frequency bioelectrical impedance analysis and dual-energy X-ray absorptiometry. Several specific (with relative missing relevant references) and minor comments detailed below prevent MS from acceptance recommendation as it is now.

Specific comments

Considerations about radioactivity exposure?

Missing relevant references regarding use/misuse of radioactivity exposure:

https://pubmed.ncbi.nlm.nih.gov/27421279/

https://pubmed.ncbi.nlm.nih.gov/26975856/

https://pubmed.ncbi.nlm.nih.gov/26261014/

https://pubmed.ncbi.nlm.nih.gov/25900295/

https://pubmed.ncbi.nlm.nih.gov/25772091/

https://pubmed.ncbi.nlm.nih.gov/25424686/

https://pubmed.ncbi.nlm.nih.gov/23135792/

Considerations about rapid weight loss before combat sports competitions?

Missing relevant references regarding rapid weight loss before combat sports competitions:

https://pubmed.ncbi.nlm.nih.gov/32182165/

Was results normality checked before opting for parametrical statistics? (l271 and 4) no mention to this study’s results normality check.

How do you specifically explain the inter-method difference comparison over upper and lower limbs (clearly shown in Figures 1 and 2)?

Minor comments

(line 72) … MRI [18,19], or… (viz., extra space)

(l84 and elsewhere throughout MS) please, do not start sentences with acronyms;

(l98) MFBIA not introduced, yet;

(l113÷7) please, move down to 2.3 Procedure;

(l145-6) … after 145 MFBIA measurements… (i.e., use subscript);

(Table 2, 2nd row) p1 (or, however, be consistent with footnote);

(l196 and elsewhere throughout MS) … in FFMarms. The… (i.e., or, however, be consistent throughout MS);

(l250) Table 2 (i.e., bold);

(l256) Figure 1 (i.e., bold);

(l256) … p=0.055… (or, however, be consistent over footnotes/captions)

(l261) Figure 2 (i.e., bold);

(l198) Figure 1(b) is the… (or, however, be consistent regarding past or present tense throughout MS)

(l234) “concentrated”?!

(l285) there is not any 36 ref in References.

Author Response

Reviewer 3

General comments

This manuscript aims at investigating differences in body composition measurements for the whole body and limb segments in elite male wrestlers of different weight classes between results of multi-frequency bioelectrical impedance analysis and dual-energy X-ray absorptiometry. Several specific (with relative missing relevant references) and minor comments detailed below prevent MS from acceptance recommendation as it is now.

Response:

Thank you for your review and suggestions. We will try our best to revise the manuscript according to your suggestions to make it more perfect.

Specific comments

Considerations about radioactivity exposure?

Missing relevant references regarding use/misuse of radioactivity exposure:

https://pubmed.ncbi.nlm.nih.gov/27421279/

 https://pubmed.ncbi.nlm.nih.gov/26975856/

https://pubmed.ncbi.nlm.nih.gov/26261014/

https://pubmed.ncbi.nlm.nih.gov/25900295/

https://pubmed.ncbi.nlm.nih.gov/25772091/

https://pubmed.ncbi.nlm.nih.gov/25424686/

https://pubmed.ncbi.nlm.nih.gov/23135792/

Response:

Thank you for your suggestions. "DXA belongs to low radiation dose. The effective dose of adult whole body examination is 0.1 to 75 μSv, which is equivalent to the equivalent dose of the subject's natural background radiation for one day or less [35]". This description has been included in the Discussion Section, Line 250-252.

  1. Bazzocchi A, Ponti F, Albisnni U, Battista G, Guglielmi G. DXA: Technical aspects and application. European Journal of Radiology. 2016; 85:1481-1492.

Considerations about rapid weight loss before combat sports competitions?

Missing relevant references regarding rapid weight loss before combat sports competitions:

https://pubmed.ncbi.nlm.nih.gov/32182165/

Response:

Thank you for your suggestions "In many sports, especially weight-based competitions, participants often suffer from acute or chronic dehydration and energy intake restriction. Weight control in abnormal ways can affect the health of athletes, especially young athletes Several studies have shown that rapid weight control is common in high school or international wrestling competitions [40, 41]. This is a very important issue for body composition measurement. In this study, because the test period was off-season, the subjects were not undergoing acute or chronic dehydration or weight control at the time of the test." We have added this narrative to the Discussion section, Line 349-355.

  1. Aloui A, Chtourou H, Souissi N. Weight reduction cycles and effects in Taekwondo. In:M. Haddad (Ed), Performance Optimization in Taekwondo: From Laboratory to Field (p. 131-136), Foster City:OMICS International.
  2. Beslija T, Cula D,, Kezic A, Tomljanovic M, Ardigo L, Dhabhi W, et al. Hight-based model for the categorization of athletes in combat sports. Applied Sport Science. 2020; 471-480.

 Was results normality checked before opting for parametrical statistics? (l271 and 4) no mention to this study’s results normality check.

Response:

Thank you for your suggestion. "Continuous variables in this study include age, weight, height, and BMI and all of them were normally distributed according to the Shapiro-Wilk tests". This statement has been added to the statistics section, line 191-192.

How do you specifically explain the inter-method difference comparison over upper and lower limbs (clearly shown in Figures 1 and 2)?

Response:

Thank you for your suggestions. "Compared with body composition measurement methods of image scanning, such as DXA or CT, bioimpedance analysis has more assumptions and requirements for body composition measurement. When bioimpedance analysis is used for single-frequency or whole-body body composition measurement, The weight of the subject needs to be input or measured. But in the BIA measurement of multiple segments, it is difficult to confirm or measure the weight of the upper or lower limbs. The current situation can only be estimated. This affects the upper or lower limbs LMST estimate is accurate". This narrative has been added to the discussion section, Line 355.362.

Minor comments

(line 72) … MRI [18,19], or… (viz., extra space)

(l84 and elsewhere throughout MS) please, do not start sentences with acronyms;

(l98) MFBIA not introduced, yet;

(l113÷7) please, move down to 2.3 Procedure;

(l145-6) … after 145 MFBIA measurements… (i.e., use subscript);

(Table 2, 2nd row) p1 (or, however, be consistent with footnote);

(l196 and elsewhere throughout MS) … in FFMarms. The… (i.e., or, however, be consistent throughout MS);

(l250) Table 2 (i.e., bold);

(l256) Figure 1 (i.e., bold);

(l256) … p=0.055… (or, however, be consistent over footnotes/captions)

(l261) Figure 2 (i.e., bold);

(l198) Figure 1(b) is the… (or, however, be consistent regarding past or present tense throughout MS)

(l234) “concentrated”?!

(l285) there is not any 36 ref in References.

Response:

Manuscripts that have been revised according to your suggestions.

Round 2

Reviewer 1 Report

Thank you, I haven't more comments

Reviewer 2 Report

The new version of the manuscript has been reviewed and it can be seen that the suggested changes have been made. I approve the publication of the manuscript.

Reviewer 3 Report

General comments

Authors addressed sufficiently all points raised by me.